# Comparative Study on Foaming Properties of Egg White with Yolk Fractions and Their Hydrolysates

**DOI:** 10.3390/foods10092238

**Published:** 2021-09-21

**Authors:** Xin Li, Yue-Meng Wang, Cheng-Feng Sun, Jian-Hao Lv, Yan-Jun Yang

**Affiliations:** 1School of Life Sciences, Yantai University, Yantai 264005, China; cfsun@ytu.edu.cn (C.-F.S.); ljh17853509809@163.com (J.-H.L.); 2School of Food and Biological Engineering, Yantai Institute of Technology, Yantai 264003, China; wangym0523@163.com; 3School of Food Science, Jiangnan University, Wuxi 214122, China; yanjunyangx@126.com

**Keywords:** foaming properties, interfacial properties, structural properties, egg yolk fractions, hydrolysates

## Abstract

As an excellent foaming agent, egg white protein (EWP) is always contaminated by egg yolk in the industrial processing, therefore, decreasing its foaming properties. The aim of this study was to simulate the industrial EWP (egg white protein with 0.5% *w/w* of egg yolk) and characterize their foaming and structural properties when hydrolyzed by two types of esterase (lipase and phospholipase A_2_). Results showed that egg yolk plasma might have been the main fraction, which led to the poor foaming properties of the contaminated egg white protein compared with egg yolk granules. After hydrolyzation, both foamability and foam stability of investigated systems thereof (egg white protein with egg yolk, egg white protein with egg yolk plasma, and egg white protein with egg yolk granules) increased significantly compared with unhydrolyzed ones. However, phospholipids A_2_ (PLP) seemed to be more effective on increasing their foaming properties as compared to those systems hydrolyzed by lipase (LP). The schematic diagrams of yolk fractions were proposed to explain the aggregation and dispersed behavior exposed in their changes of structures after hydrolysis, suggesting the aggregated effects of LP on yolk plasma and destructive effects of PLP on yolk granules, which may directly influence their foaming properties.

## 1. Introduction

Foam is normally defined as a two-phase colloidal structure that is present in a non-equilibrium state containing a continuous phase (usually water) in which gas (usually air) is suspended as bubbles or gas cells [1,2,3]. Foamability and foam stability are important subjects within food colloids since foams are highly packed by large numbers of bubbles that form an interconnected structure in a wide range of food products, for example, in bread, ice cream, cakes, mousse, etc. [4,5]. Therefore, these two indexes (foamability and foam stability) are key standards for determining the quality of foamed products. Apart from that, egg white is also popular in other industries, including cheese, meat, wine, beer and oil for other properties. Egg products (liquid egg, white powder, yolk powder and egg powder) are popular in China and thus meaningful for investing.

Egg white protein (EWP) is a classic foaming material in foods such as meringues and cake batters [6]. EWP is obtained by completely separating egg white and egg yolk from eggs. Egg white has many functional properties, including gelling, coagulating, keeping water and antioxidant properties. As an excellent foaming agent, the foaming properties of EWP are influenced by many factors, including egg freshness, pH values, protein concentration, shearing conditions and contaminative egg yolk [7]. Yolk contamination is the major factor that can reduce or destroy the foaming properties of EWP in food industries, especially bakery factories [8]. A low percentage of egg yolk was demonstrated to significantly reduce the foamability of EWP [9,10]. However, egg white cannot be completely separated from egg yolk in egg processing companies, including with current advanced egg cracking and breaking technologies [11,12]. Therefore, it is meaningful to research methods in the improvement of industrial EWP (i.e., egg white protein contaminated with egg yolk) and their mechanisms. Although physical methods such as UV irradiation fluoroscopy, visible light and near-infrared transmission fluoroscopy, etc. [13,14,15] were employed to increase the foaming properties of egg yolk or whole egg liquids, they could not eliminate the egg yolk in egg white. In order to decrease or eliminate the negative effects of egg yolk on the foaming properties of egg white, the components of egg yolk should be deeply understood. It is known that yolk can be easily separated into two fractions: the supernatant, called yolk plasma, and the precipitate, called yolk granules, as reported by others [16,17]. Yolk plasma is composed mainly of low-density lipoproteins (LDLs) and soluble proteins (livetins), while yolk granules mainly constitute high-density lipoproteins (HDLs), phosvitin and a small amount of LDLs [18].

Foaming and structural properties of egg white protein with or without yolk or yolk fractions (plasma and granules) were investigated before and after hydrolysis by lipase and phospholipase A_2_. It is meaningful to determine the effects of esterase treatments on foaming characteristics of industrial egg white protein (egg white contaminated with a low percentage of egg yolk). Therefore, the objective of this study was to compare the foaming and structural properties of industrial egg white protein (egg white contaminated with 0.5% egg yolk, *w/w*) being hydrolyzed by lipase and phospholipase A_2_, and to determine their positive effects on foaming properties of industrial egg white protein via structural changes. Furthermore, lipase and phospholipase A_2_ were employed in this article in order to enhance and expand their applications scope in food systems, especially in aerated systems.

## 2. Materials and Methods

### 2.1. Materials

Fresh hen eggs (less than 5 days after laying) were purchased from a local supermarket in Wuxi, Jiangsu Province, China. Lipase (LP, 3000 U/g) was purchased from Sinopharm Chemical Reagent Co., Ltd. (Shanghai, China) and phospholipase A_2_ (PLP, 5000 U/mL) was kindly handsealed by Youpuke Biological Technology Co., Ltd. (Wuxi, China). Sodium dihydrogen phosphate, di-sodium hydrogen phosphate and sodium azide were purchased from Sigma-Aldrich (Beijing, China). Water purified by treatment with a Milli-Q apparatus (Millipore, Bedford, UK), with a resistivity not less than 18.2 MΩ cm at 25 °C, was used for the preparation of phosphate buffer. The latter was used as the solvent throughout the experiments with addition of 0.02 wt% sodium azide as a bactericide. All the other chemicals and reagents used in this study were of analytical grade.

### 2.2. Preparation of Samples

#### 2.2.1. Preparation of Egg White Protein Dispersion

Egg white was manually extracted from the yolk of freshly purchased eggs immediately upon arrival in the lab and then homogenized under magnetic stirring (500 rpm speed) for 2 h, as reported previously [6]. No further purification of the egg white protein dispersion (EW) was performed.

#### 2.2.2. Preparation of Egg Yolk and Its Fractions

According to our previous method [9], the yolk membrane was carefully punctured to effuse the unspoiled egg yolk, gently homogenized at 4 °C for 2 h and then egg yolk (EY) dispersion was achieved. In order to perform its fractions, egg yolk was diluted with water at a ratio of 1:1.5 (*v/v*) and adjusted to pH 7.0 with 1 N NaOH. After equilibrating overnight at 4 °C, dispersions were centrifuged 45 min (10,000× *g*) at 4 °C. Then, egg yolk was set apart into egg yolk fractions including plasma (the supernatant) and granules (the precipitate) (Appendix A).

#### 2.2.3. Preparation of Hydrolysates

Hydrolysates of egg yolk and its fractions were prepared by enzymatic hydrolysis, which was conducted as described by Xing Fu [19] using the pH-stat method. Plasma and granules were dissolved in deionized water to prepare their dispersions (EP and EG) for 1:9 (*w/w*) and 9:1 (*w/w*), respectively. Therefore, the yield of plasma and granules corresponded to their preparation in yolk. Lipase (LP) and phospholipase A_2_ (PLP) were dissolved in 10 mM phosphate buffer (PBS) at pH 8.0 in advance. EP and EG were heated to 45 °C quickly and hydrolyzed by LP and PLP at 100 U/g (yolk basis) and 500 U/g (yolk basis) for 2 h in a water bath (under quiescent conditions) at pH 8.0, where enzyme reaction had been completed (not shown), followed by cooling in ice water. Thus, hydrolysates of EP and EG were obtained and called LEP (EP hydrolyzed by LP), PEP (EP hydrolyzed by LP), LEG (EG hydrolyzed by PLP) and PEG (EG hydrolyzed by PLP). In order to make a comparison, hydrolysates of EY (LEY, PEY) were hydrolyzed by LP and PLP similarly for further tests.

### 2.3. Foaming Characterization

#### 2.3.1. Foam Preparation

In order to compare the foaming properties of egg yolk fractions and their hydrolysates, all yolk (yolk fractions) dispersions prepared in Section 2.2 were combined with EW (total quantity was 30 g) and whipped in 150 mL beakers for 2 min, using hand-held electric whisks (N20D, Netmego, FoShan, Guangdong, China) under atmospheric pressure. Thus, different foam systems were obtained to describe their foaming characterizations.

#### 2.3.2. Foam Capability (FC)

Foam samples were all transferred to cylindrical containers, and a spatula was used to level the top of the foams in order to achieve uniform and plane surfaces to record the foam volume precisely. The foam capability (FC) could be calculated by Equation (1):FC (%) = (m_i_ − m_f_)/m_f_ × 100(1)
where m_i_ represents the mass of unwhipped dispersions and m_f_ represents the mass of the resulting whipped dispersions (foams) with the same volume of m_i_.

#### 2.3.3. Foam Stability (FS)

Foam stability (FS) was expressed by the mass of drained fluid from the lamella after the whipped foams were stored for 30 min in room temperature and could be calculated by Equation (2):FS (%) = m_2_/m_1_ × 100(2)
where m_1_ represents the initial mass of foam and m_2_ represents the mass of drained liquid.

### 2.4. Particle Size Distribution

The particle size distribution (PSD) under different treatments was measured by dynamic light scattering via a Zetasizer Nano-ZS (Malvern instruments, Worcestershire, UK) at 25 °C. The refractive index of proteins and aqueous phase was set at 1.45 and 1.33, respectively. Each sample was measured in triplicate.

### 2.5. Measurement of Zeta Potential

Zeta-potentials of sample dispersions were measured in the standard folded capillary electrophoresis cells, using a Zetasizer Nano ZS instruments (Malvern instruments, Worcestershire, UK). Before measurement, sample dispersions were diluted to 0.005% protein particle concentration using Milli-Q water. Each individual zeta-potential data point was reported as the average and standard deviation of at least five reported readings made on triplicate samples.

### 2.6. Confocal Laser Scanning Microscopy (CLSM)

Confocal laser scanning microscopy (CLSM) of the sample dispersions before and after hydrolyzing were imaged via a Zeiss LSM 710 confocal microscope (Carl Zeiss MicroImaging GmbH, Jena, Germany), where sample systems were imaged after mixing with 0.1 mL of 1.0% (*w/v*) Rhodamine B protein stain (excitation 568 nm) and 0.1 mL of 1.0% (*w/v*) Nile red oil stain (excitation 488 nm). The samples were observed at room temperature (25 °C ± 1 °C) using × 63 objective [20].

### 2.7. Scanning Electron Microscopy (SEM)

For SEM analysis, sample dispersions were imaged via a scanning electron microscope (SU8220, Hitachi, Japan), as described by others [21,22]. Briefly, samples were lyophilized, and the dried powder was sprinkled onto a two-sided adhesive tape and then coated with a thin layer of gold to analyze.

### 2.8. Statistical Analysis

All measurements were conducted at least in triplicate, and SPSS version 19.0 was used for statistical analysis of means and standard deviations. Significant differences between samples were determined by one-way analysis of variance (ANOVA) with Duncan’s adjustment performed, and the level of confidence was 95%.

## 3. Results

### 3.1. Foaming Properties

Foaming properties of egg white protein influenced by yolk fractions and their hydrolysates are shown in Figure 1. Compared with EW itself, foam capability (FC) and foam stability (FS) of egg white dispersions with yolk and plasma had an intuitively significant reduction, which meant yolk and plasma might compete with egg white at the air–water interface. However, as another fraction of yolk, granules did not harm the foam ability of egg white proteins as shown in Figure 1. This result may be ascribed to the high protein concentration and robust network structure in granules. As described by Kiosseoglou [23], phosvitin in granules exhibited in the form of HDL-phosvitin complexes can contribute to the powerful dense structure through phosphocalcic bridges and stabilize the peculiar hydrophobic cavum, which can adsorb and retain air. With yolk hydrolysates, a rapid increase in FC was observed for EW compared with the unhydrolyzed ones. Thus, foam capacities of hydrolysates of yolk, plasma and granules were higher than themselves. This may be attributed to their faster adsorbing ratio to the A/W interface after hydrolysis by virtue of the smaller size and lower concentration of lipids [16,17]. In addition, the effect on increasing foaming properties of PLP is better than LP shown in Figure 1. However, their abilities of improving the foam stability seemed to be different. An advantage was exhibited for yolk fraction hydrolysates when hydrolyzed by PLP in improving FS of EW. With dispersions hydrolyzed by LP, the FS of EW could not be improved compared to those with dispersions hydrolyzed by PLP. Thus, in general, this confirmed that the foam stabilities of EW + yolk fractions (PLP) systems were higher than those of the EW + yolk fractions (LP) systems, in this case due to different hydrolysis mechanisms, which should be demonstrated later.

### 3.2. Interfacial Properties

#### 3.2.1. Particle Size

Results of particle size distribution are shown in Figure 2. As can be seen in Figure 2a, EY seemed to have a smaller mean hydrodynamic diameter since the particle size distribution ranged from right to left after different treatments (heated, hydrolyzed by LP and hydrolyzed by PLP). However, the particle distribution seemed to shift to the lowest sizes when NEY (natural egg yolk) was hydrolyzed by PLP in Figure 2a, which may indicate that the most destructive structure can be observed in this condition. Figure 2b compares the size distribution of EP that stabilized after different treatments. As seen, there are not significant differences for EP in Figure 2b after heating or hydrolysis by LP compared with EG, as shown in Figure 2b, except for the similar small peaks below 100 nm that appeared after treatments. Therefore, the smaller peak probably represents EP that somehow joined the protein hydrolysis process. In contrast, a great shift of size distribution for NEG happened after our processes, which means the greatest changes may occur in EG for yolk fractions after treatments. In addition, all dispersions hydrolyzed by PLP had a smaller mean hydrodynamic diameter compared with the unhydrolyzed ones, with a particle size distribution showing the only prominent peak in the region of 100–1000 nm. Overall, it is seen that hydrolysis contributed to a size decrease in yolk fractions, whereas PLP seemed more destructive than LP, owing to its effects on their protein structure [24,25].

#### 3.2.2. Zeta Potential

As previously reported by Gao et al. [26], hydrolysis of yolk fractions showed an apparent effect on the zeta potential. Figure 3 shows the zeta potential of sample dispersions under different treatments. Compared to the untreated yolk fractions, there is no significant difference for heated ones, while the absolute zeta potential value of hydrolyzed ones increased. These results suggest that the exposed charged groups on the surface of yolk fractions were increased after hydrolysis, regardless of hydrolysis by lipase or phospholipase [27]. Results of zeta potential (absolute values) showed an increase after yolk fractions hydrolyzed by lipase and phospholipase A_2_. However, the absolute value of zeta potential for yolk fractions hydrolyzed by LP was higher than that of hydrolyzed by PLP, which may be attributed to the different effects of lipase on yolk structures. The zeta potential of LEG (egg granule hydrolyzed by lipase) approximately reached −35 mV, and this phenomenon may be due to the fact that the higher surface charge will be possessed for yolk granules after hydrolysis by PLP compared to NEG (natural egg granule), although different extents will happen for the groups exposed after different hydrolyses [28].

### 3.3. Structural Properties

#### 3.3.1. CLSM Analysis

The CLSM images of sample dispersions under different treatments (natural yolk fractions, heated yolk fractions, yolk fractions hydrolyzed by LP and yolk fractions hydrolyzed by PLP) are presented in Figure 4, the lipid components marked by red are surrounded by proteins labeled blue. For the fresh and natural yolk fraction dispersions in Figure 4a, lipid components are spaced evenly in the view, especially yolk and plasma dispersions. After heating, it is obvious in Figure 4b that lipid components were aggregated in all samples, while there is not considerable change for the protein components, except in heated granules, which may be attributed the coalescence during the freeze-drying step before testing [29]. Compared with the spherical structure in Figure 4a, the circle structure is visible in Figure 4b in yolk plasma images, indicating that the lipids were aggregated in a large scale after heating [30].

As can be seen, both lipid and protein components in the image are relatively large in Figure 4c. These results revealed that after hydrolysis by LP, yolk fractions can coalescence to a certain extent. Furthermore, the large amount of lipids in images has favored the formation of free fat, which was possibly affected by two steps: yolk fractions hydrolyzed by LP and the free fat aggregated after exposure during the hydrolysis in yolk fractions [31]. However, we also observed that the polygonal shape of components in yolk fraction images had pores and channels that contain large amount of protein and lipids [32], indicating that the aggregation step was more crucial than the dispersing step for yolk fractions after hydrolysis by LP. In Figure 4d, the most dispersed and uniform visions were observed, which indicated the structures of yolk fractions (yolk, plasma and granules) may be completely destroyed after hydrolysis by PLP. In other words, components in plasma and granules may respond to PLP. It is tempting to propose that both low-density lipoprotein (LDL) in plasma and high-density lipoprotein (HDL) in granules seemed to be hydrolyzed by PLP since LDL mainly exists in yolk plasma and HDL in yolk granules [9].

#### 3.3.2. SEM Analysis

SEM images of sample dispersions, including natural yolk fractions (NEY/NEP/NEG), yolk fractions hydrolyzed by lipase (LEY/LEP/LEG) and yolk fractions hydrolyzed by phospholipase A_2_ (PEY/PEP/PEG), are shown in Figure 5. For natural yolk fraction samples (NEY/NEP/NEG), compact and dense structure can be observed, while fragmentized clumps and a looser structure with many folds appeared after hydrolysis by LP and PLP. These results indicated that both lipase and phospholipase A_2_ may effectively act on the complex structures in yolk (both plasma and granules). However, partially aggregated particles were visible, which might have been caused by the heat treatments during hydrolysis [33,34]. As can be observed, there are many roughened pores and pitted surfaces that appeared in LEG, which indicates that LP acted on the inside of yolk granules instead of acting on the surface. The most roughened structures that happened in yolk fractions hydrolyzed by PLP was due to the strong effects of PLP, even if the structures of yolk fractions were not completely in pieces in the SEM images [35].

### 3.4. Schematic Illustration for the Hydrolysates

Based on our results, both yolk plasma and yolk granules had undergone the structural changes after hydrolyzed by LP and PLP. Therefore, the schematic diagrams of yolk fractions hydrolyzed by LP and PLP were proposed in Figure 6 (yolk plasma) and Figure 7 (yolk granules) in order to explain more directly the aggregation or dispersed behavior of yolk fractions after hydrolysis.

As reported, the main constituents in yolk plasma were livetin and low-density lipoprotein (LDL), which includes phospholipids, cholesterol, apoproteins and lipids [36,37]. From the perspective of yolk plasma, the structure was considerably looser when hydrolyzed by PLP versus LP. During the hydrolysis of LP, free lipids were exposed and hydrolyzed, while the LDL were partially destroyed and aggregated due to the enhancement of electrostatic interactions between egg yolk proteins, as shown in Figure 3. Structural changes occurred largely on the surface of the yolk plasma, leading to the dispersion of LDL (Figure 5 LEP, egg plasma hydrolyzed by lipase). When hydrolyzed by PLP, the LDL was not totally destroyed but caused aggregates to crack or expand to form small particles on the surface (Figure 5 PEP, egg plasma hydrolyzed by phospholipase). However, the inner structural changes might have happened for the yolk granules. In Figure 7, we have attempted to describe the structural changes more accurately by proposing the schematic diagram of yolk granules after hydrolysis by LP and PLP. Previous studies also reported that the main constituents in yolk granules were high-density lipoprotein (HDL), low-density lipoprotein (LDL) and phosvitin, which were combined with a calcium phosphate bridge [38]. After hydrolysis by LP, the surface coverages seemed not to be completed since lipids in LDL were easier affect than that in HDL. Thus, the honeycomb structure in yolk granules could be seen after hydrolysis by LP (Figure 5 LEG). It is noteworthy for granules that phospholipids were present in both HDL and LDL, as reported by others [39,40,41]. Phospholipase were confirmed to mainly act on phospholipids [42,43]. It can clearly be seen in Figure 4d that the structure of yolk fractions was loosest after hydrolysis by PLP. From the point of view of yolk granules, apoproteins in HDL could not be destroyed, while parts of phospholipids were hydrolyzed by PLP, leading to pieces of granules in Figure 5 PEG. In general, PLP increases the denaturation contents of phospholipids in yolk granules and delays the formation of aggregates, contributing to the complete destruction of granule structures [40].

## 4. Conclusions

Foaming and structural properties of egg yolk fractions (including yolk itself, yolk plasma and yolk granules) and their hydrolysates (hydrolyzed by lipase and phospholipase A_2_) were examined. EWP was taken as a standard. Our findings seem to validate the fact that both lipase and phospholipase A_2_ can increase the foaming properties of yolk fractions, including higher than EWP itself when combined with EWP. However, two types of enzyme showed different mechanisms on the effects on foaming properties of yolk fractions. Compared with LP, higher foamability and foam stability were obtained when hydrolyzed by PLP, which may be attributed to the physicochemical and structural properties of the hydrolysates. Measurements of structural properties seemed to support the idea that loose structures of yolk fractions were beneficial in increasing their foaming properties. Therefore, the fundamental insights of this study may pave a way for improving the foaming properties of industrial egg white (i.e., egg white contaminated by egg yolk) by hydrolyzing sample dispersions. In another aspect, it also provided an idea to generate new types of food foams by retrofitting yolk fractions.

## Figures and Tables

**Figure 1 foods-10-02238-f001:**
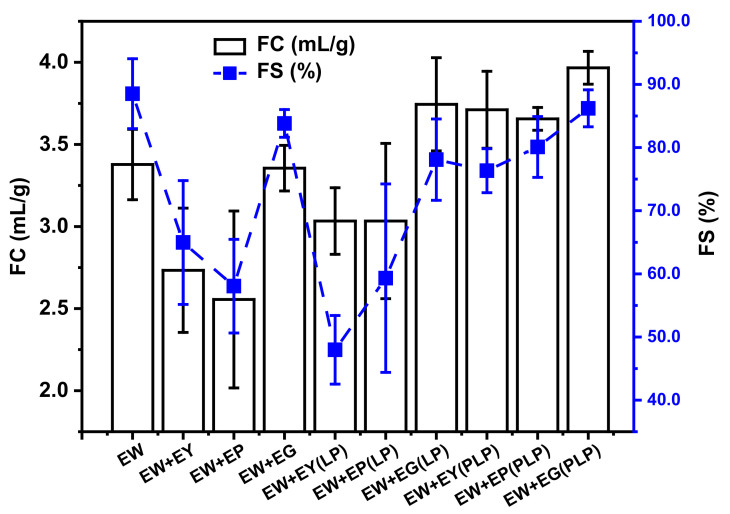
Foam capability (FC) and foam stability (FS) of egg white (EW) with egg yolk fractions, including egg yolk (EY), yolk plasma (EP) and yolk granule (EG) and their hydrolysates (hydrolyzed by lipase LP or phospholipase PLP).

**Figure 2 foods-10-02238-f002:**
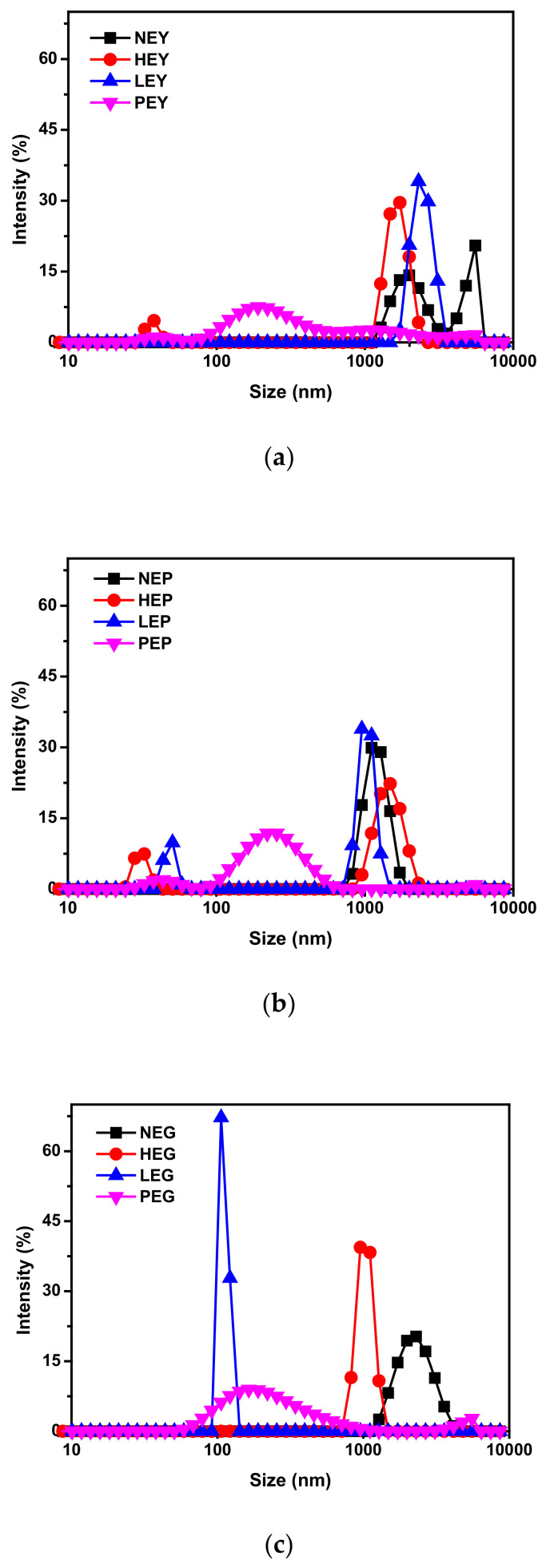
Particle size distribution of sample dispersions under different treatments. (**a**) NEY, HEY, LEY and PEY (natural egg yolk, heated egg yolk, egg yolk hydrolyzed by lipids and egg yolk hydrolyzed by phospholipase, respectively). (**b**) NEP, HEP, LEP and PEP (natural egg yolk plasma, heated egg yolk plasma, egg yolk plasma hydrolyzed by lipids and egg yolk plasma hydrolyzed by phospholipase, respectively). (**c**) NEG, HEG, LEG and PEG (natural egg yolk granule, heated egg yolk granule, egg yolk granule hydrolyzed by lipids and egg yolk granule hydrolyzed by phospholipase, respectively).

**Figure 3 foods-10-02238-f003:**
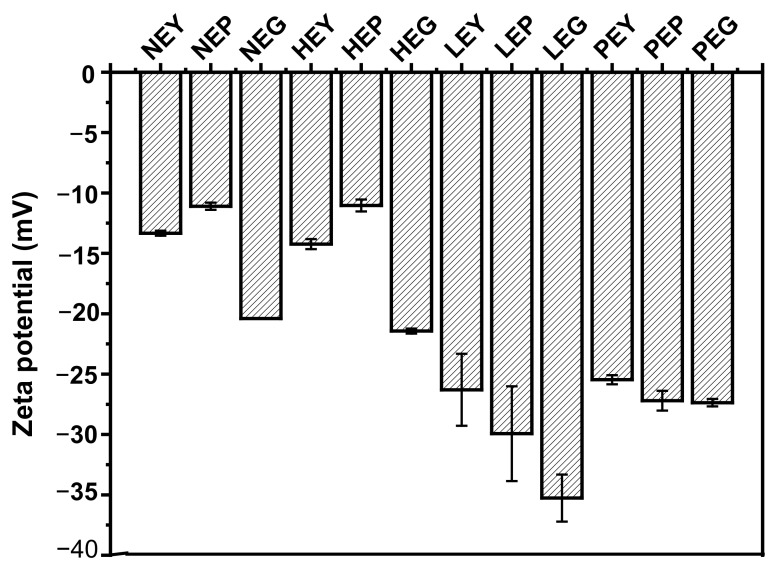
Zeta potential of egg yolk fractions (egg yolk, plasma and granule) under different treatments (heat treatment, hydrolyzed by lipase and hydrolyzed by phospholipase). NEY, NEP and NEG (natural egg yolk, natural egg plasma and natural egg granule, respectively); HEY, HEP and HEG (heated egg yolk, heated egg plasma and heated egg granule, respectively); LEY, LEP and LEG (egg yolk hydrolyzed by lipase, egg plasma hydrolyzed by lipase and egg granule hydrolyzed by lipase, respectively); PEY, PEP and PEG (egg yolk hydrolyzed by phospholipase, egg plasma hydrolyzed by phospholipase and egg granule hydrolyzed by phospholipase, respectively).

**Figure 4 foods-10-02238-f004:**
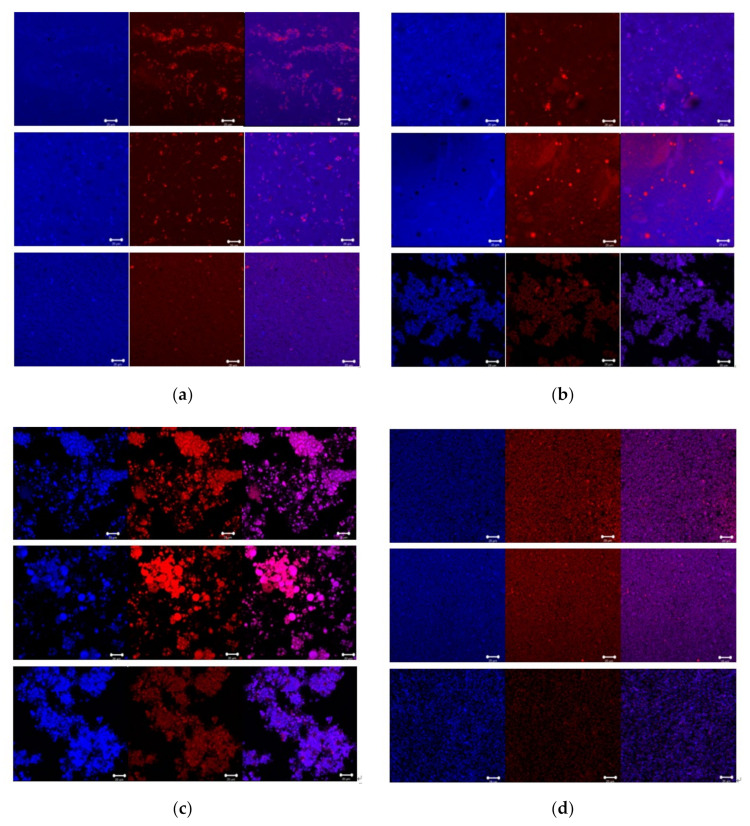
Confocal images of sample dispersions under different treatments. (**a**) NEY, NEP and NEG (natural egg yolk, natural egg plasma and natural egg granule, respectively). (**b**) HEY, HEP and HEG (heated egg yolk, heated egg plasma and heated egg granule, respectively). (**c**) LEY, LEP and LEG (egg yolk hydrolyzed by lipase, egg plasma hydrolyzed by lipase and egg granule hydrolyzed by lipase, respectively). (**d**) PEY, PEP and PEG (egg yolk hydrolyzed by phospholipase, egg plasma hydrolyzed by phospholipase and egg granule hydrolyzed by phospholipase, respectively). Scale bars in images mean 20 μm.

**Figure 5 foods-10-02238-f005:**
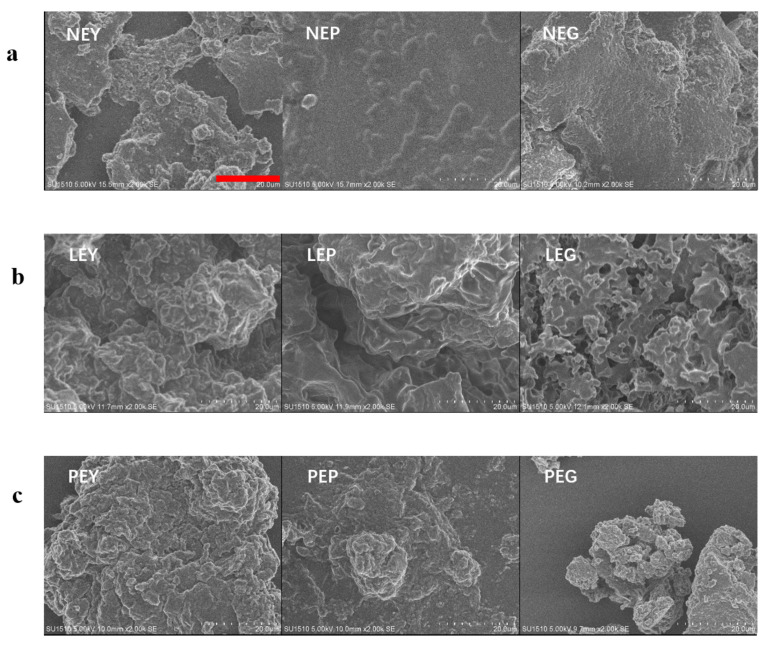
SEM images of sample dispersions under different treatments. (**a**) NEY, NEP and NEG (natural egg yolk, natural egg plasma and natural egg granule, respectively). (**b**) LEY, LEP and LEG (egg yolk hydrolyzed by lipase, egg plasma hydrolyzed by lipase and egg granule hydrolyzed by lipase, respectively). (**c**) PEY, PEP and PEG (egg yolk hydrolyzed by phospholipase, egg plasma hydrolyzed by phospholipase and egg granule hydrolyzed by phospholipase, respectively). The red scale bar in the image means 20.0 μm.

**Figure 6 foods-10-02238-f006:**
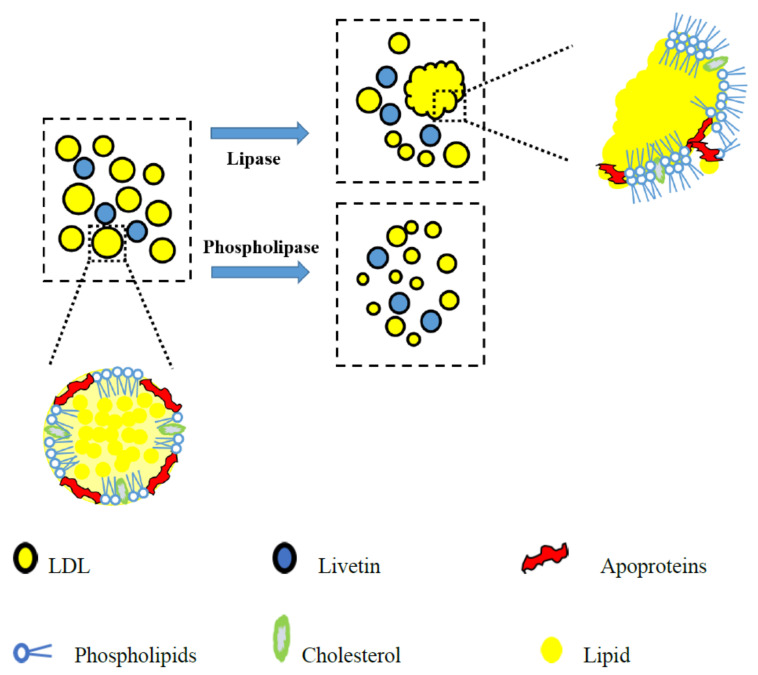
Proposed schematic diagram of egg yolk plasma hydrolyzed by lipase and phospholipase.

**Figure 7 foods-10-02238-f007:**
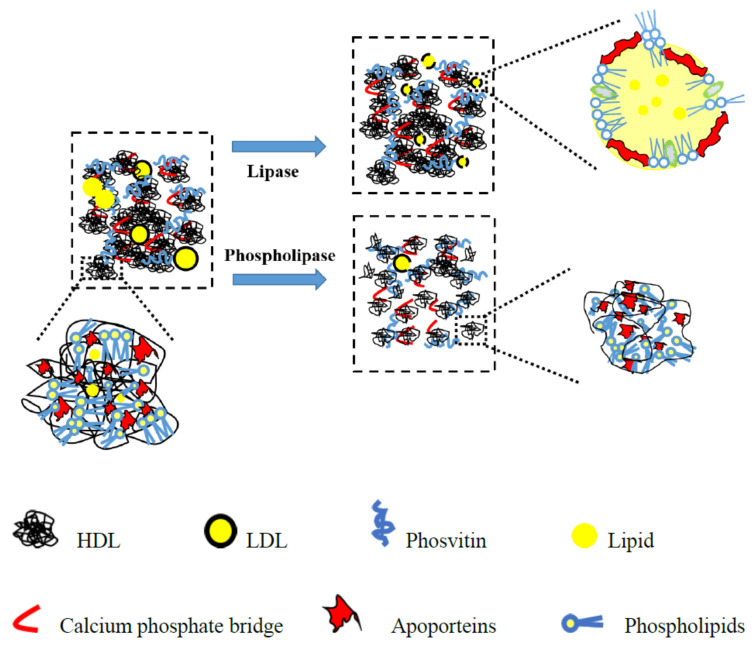
Proposed schematic diagram of egg yolk granules hydrolyzed by lipase and phospholipase.

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
