# Peer review of "Comparative Study on Foaming Properties of Egg White with Yolk Fractions and Their Hydrolysates"

_foods, 2021, doi:10.3390/foods10092238_

Round 1

Reviewer 1 Report

The aim of the research was to compare the foaming and structural properties of table egg white protein being hydrolyzed by lipase and phospholipase A2. The number of eggs must be clearly stated in the Abstract and Materials and Methods chapters. The test methods used are correct. The chapter "Introduction" requires some additions. The discussion is well carried out and exhausting. References well chosen but must be revision according to instructions for authors. The paper requires additions and corrections. The list of proposed changes is given below:

General comments:

Please prepare the article according to the instructions for the authors.

For affiliates, the first name and surname initials for each co-author (also Yue-Meng Wang and Yan-Jun Yang) of the article should be provided, the same as given in the "Author Contributions" chapter.

Missing chapters "Author Contributions", "Funding" Data Availability Statement, “Conflicts of Interest” must be added

The references chapter must be prepared in accordance with the instructions: abbreviated name journal in italic, publication year in bold, volumen number in italic

Detailed comments:

L34 Please add something about the use of egg white in the cheese industry (ripened cheese), meat industry, wine and beer production, oil industry. How is the production of egg products (liquid egg, white powder, yolk powder, egg powder) in China.

L37 please add information about other functional properties of egg white: apart from creating and stabilizing foam, these include: gelling, coagulating, keeping water. Egg white also has antioxidant properties

L46 + what is the effect of disinfecting table eggs on the freshness and quality of the liquid egg content. What is the effect of egg storage length and temperature on egg foaming capacity? The importance of de-saccharification of egg white in the production of egg products on foaming of egg white

L70 "Fresh hen eggs" is when from laying eggs? 2-3 days or 2-3 weeks. On the packages was the use-by dates.

L70 when done post-purchase research how eggs were stored pending testing

L170 [16,17] instead of [17,16]

L184 “Particle” with capital letter

Figure 2 - larger size, one on one level, larger font for legend

Conclusions should refer to the title of the article and the purpose of the research, Lack information about foaming properties of egg white?

References as instructed, no [J], no double spaces, but with abbreviated name journal in italic, publication year in bold, volume number in italic.

Reviewer 2 Report

Remarkably defect free manuscript (and figures). Supplementary information is also excellent quality. cogent and clear elucidation of aqueous foam lamellar behaviour and surface coverage explained by methods selected.

NB: Interfacial tension measurements would perhaps have further enhanced this work? It is worth thinking about "tensiometry" (surface pressure, surface rheology) as a a probe of structure/function, in future work

Error: line 75 typo: handseled should be written as handsealed
